# Assessing the Hazard of Deep-Seated Rock Slope Instability through the Description of Potential Failure Scenarios, Cross-Validated Using Several Remote Sensing and Monitoring Techniques

Charlotte Wolff [1,*], Michel Jaboyedoff [1], Li Fei [1], Andrea Pedrazzini [2], Marc-Henri Derron [1], Carlo Rivolta [3] and Véronique Merrien-Soukatchoff [1,4]

[1] Risk Analysis Group, Institute of Earth Science, University of Lausanne, 1015 Lausanne, Switzerland; michel.jaboyedoff@unil.ch (M.J.); li.fei@unil.ch (L.F.); marc-henri.derron@unil.ch (M.-H.D.); veronique.merrien@lecnam.net (V.M.-S.)
[2] Sezione Forestale—Repubblica e Cantone Ticino, 6500 Bellinzona, Switzerland; andrea.pedrazzini2@ti.ch
[3] Ellegi Srl—Via Bandello 5, 20123 Milano, Italy; carlo.rivolta@lisalab.com
[4] Laboratoire Géomatique & Foncier, Conservatoire National des Arts et Métiers, 75013 Paris, France
[*] Correspondence: charlotte.wolff@unil.ch

**Abstract:** Foreseeing the failure of important unstable volumes is a major concern in the Alps, especially due to the presence of people and infrastructures in the valleys. The use of monitoring and remote sensing techniques is aimed at detecting potential instabilities and the combination of several techniques permits the cross-validation of the detected movements. Supplemented with field mapping and structural analysis, it is possible to define possible scenarios of rupture in terms of volume, mechanisms of failure and susceptibility. A combined observation strategy was applied to the study of major instability located in the Ticinese Alps (Switzerland), Cima del Simano, where the monitoring started in 2006 with the measurement of opened cracks with extensometers. Since 2021, the monitoring has been completed by LiDAR, satellite and GB-InSAR observations and structural analysis. Here, slow but constant movements of about 7 mm/yr were detected along with rockfall activities near the Simano summit. Eight failure scenarios of various sizes ranging from $2.3 \times 10^5$ m$^3$ to $51 \times 10^6$ m$^3$, various mechanisms (toppling, planar, wedge and circular sliding) and various occurrence probabilities were defined based on the topography and the monitoring results and by applying a Slope Local Base Level (SLBL) algorithm. Weather acquisition campaigns by means of thermologgers were also conducted to suggest possible causes that lead to the observed movements and to suggest the evolution of the instabilities with actual and future climate changes.

**Keywords:** instability monitoring; LiDAR; InSAR; extensometer; SLBL; rockslide; rupture scenario; susceptibility assessment

## 1. Introduction

Steep and narrow valleys are common in the Alpine mountains, often crossed by main roads, where infrastructures and communities are settled. Studies mapping potential slope instabilities and valleys exposed to the propagation of those instabilities are numerous [1–3]. Other studies highlight the possible mechanisms of the rupture of those instabilities, especially in the case of deep-seated landslides involving important volumes [4,5], through structural analysis [6–8]. Their destabilization is often controlled by the topography, the geology and the structural inheritance [9–12]. The Ticino canton in the southern Swiss Alps is particularly affected by rock instabilities and landslides [13–15] especially after important precipitations like those that occurred in July 2021 [16]. The Monte Crenone landslide and the triggered "Buzza di Biasca" events are among the oldest historical natural hazard events reported in the Alps [17,18].

The use of remote sensing techniques is very convenient to monitor and analyze slope instabilities. Light detection and ranging (LiDAR) or Laser scanner are commonly used to characterize rockfall events in vertical cliffs [19–21] as well as the main discontinuity sets constraining sliding or toppling movements [22,23]. The Global Navigation Satellite System (GNSS) [24,25], satellite Interferometric Synthetic Aperture Radar (InSAR) and Ground-Based InSAR (GB-InSAR) [26–28] techniques provide worthy information on slow and/or deep-seated rockslide movements for integration in early-warning systems (EWS, [29,30]). Doppler radar is another monitoring technique growing in popularity for rockfalls detection and tracking [31–33]. It is also common to combine several monitoring techniques to confirm movements and better characterize the different types of instabilities and movements occurring on steep cliffs and slopes [3,34,35].

Decisions on long-term monitoring, EWS and mitigation measures are often based on the description of scenarios for the geometry and the volume of unstable compartments defined by a specific failure mode. Remote sensing monitoring can thus help in better describing those scenarios [36].

Cima del Simano instability is a summit constituted by gneisses located in the same valley as the one where the Monte Crenone landslide occurred in 1513 [37]. It presents one main 500 m-long open fracture up to a width of 10 m and several smaller fractures of various sizes closer to its crest. Signs of movements, such as toppling blocks and fresh soil in the fractures could be found at the front of the summit. Due to its remote and high-altitude location, the monitoring was conducted by combining five different remote sensing and monitoring techniques, which are extensometers, LiDAR, the drone Structure from Motion (SfM), satellite InSAR and GB-InSAR; all these techniques are complementary, catching various pieces of information on different types of movements.

From the results of the monitoring campaign, coupled with a structural analysis of the main discontinuity sets measured in the field and on Point Clouds (PCs), height different scenarios of potential large rock instabilities have been proposed with different volumes, extents and failure modes. This study shows that the confrontation of geomorphic analysis and several independent displacement measurements permitted a cross-validation of these proposed scenarios and gives an idea on their probability of occurrence.

## 2. Description of Case Study

In the Ticino Canton (Switzerland), landslides are recurrent natural hazard phenomena occurring due to very steep slopes induced by glaciers, rapid uplift [38] and heavy rainfalls. Many studies map and describe those threats [14,39]. One of the present concerns is the Cima del Simano Mountain in the Blenio Valley, near the village of Acquarossa (Figure 1a). The area of instability is part of the Simano Nappes [40] and is made up of an orthogneiss with the main schistosity oriented, on average, to 20°/035° [41]. The top, reaching an altitude of 2500 m, displays several fractures of various sizes. A 500 m-long major opened crack is located at 100 m toward the southeast behind the crest and identified as '001' in Figure 1b. Nevertheless, numerous other cracks of a smaller length and smaller aperture are also visible closer to the crest line and some delineate clear depressions (Figure 1c,d). They follow two main lineaments oriented at 040° and 157° towards north, respectively mapped in white and black in Figure 1c. A regional fault also crosses the mountain perpendicularly to the crest. A major depression named D1 in Figure 1c is visible in the field and is covered by the debris of rocks (Figure 2).

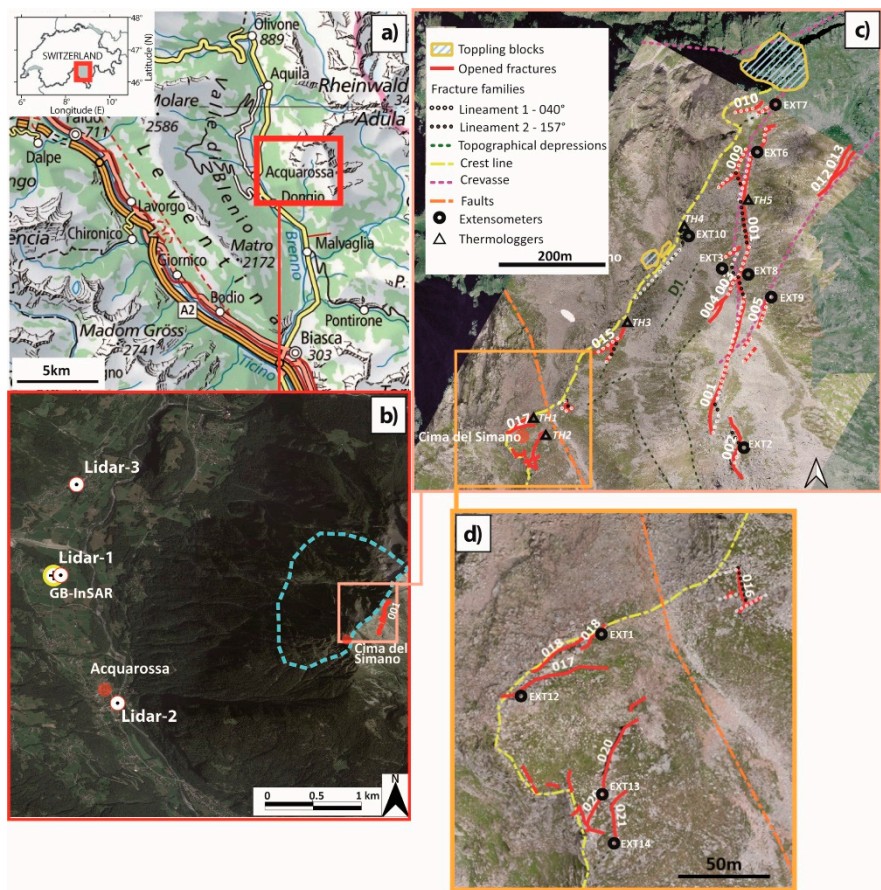

**Figure 1.** Characterization of the case study with the location of the remote sensing devices installation. (**a**) Localization. (**b**) LiDAR and GB-InSAR installed in the Valley, distance 3.5 km from the mountain peak. The area illuminated by the GB-InSAR is delimitated in blue (**c**). (**d**) Mapping on the orthophoto from SfM photogrammetry of the fractures, faults and depressions on top of Cima del Simano and extensometers and thermometers installed in the fractures.

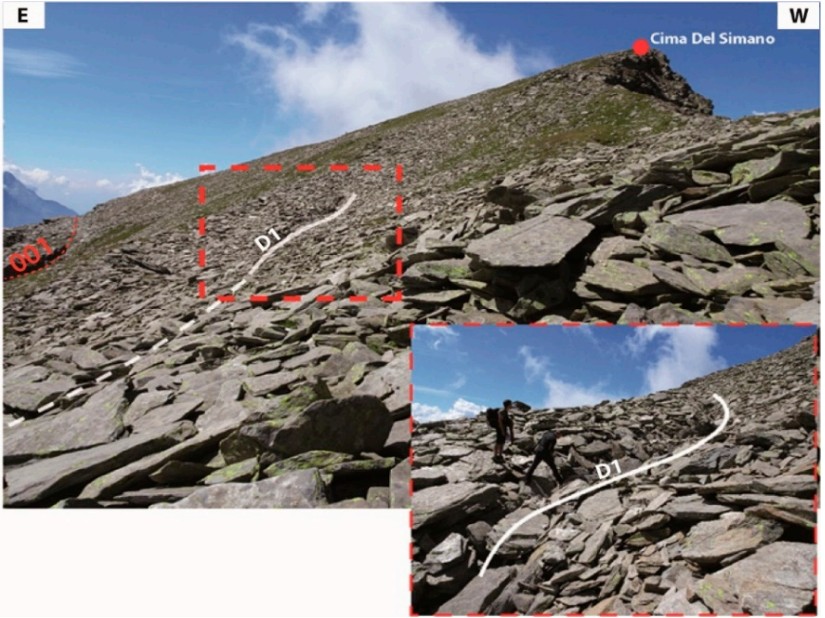

**Figure 2.** Photo of the depression D1 mapped in Figure 1c.

The southeastern wall of the major fracture '001' is nearly vertical. The opposite wall is only visible in the northern part, near fractures '003' and '004', and covered by debris in the southern part, near fracture '002'. The topography facing the southeastern wall slopes at a 30° angle toward the southeast (Figure 3a). This fracture seems to be the result of a gravitational movement of the northwestern block moving downward (Figure 3b). Signs of recent movements and instabilities are revealed close to the crest in some of the fractures, such as the presence of soil disturbance found in the fracture '018' (Figure 3c) or the presence of unstable blocks along the crest that are clearly toppling (Figure 3d).

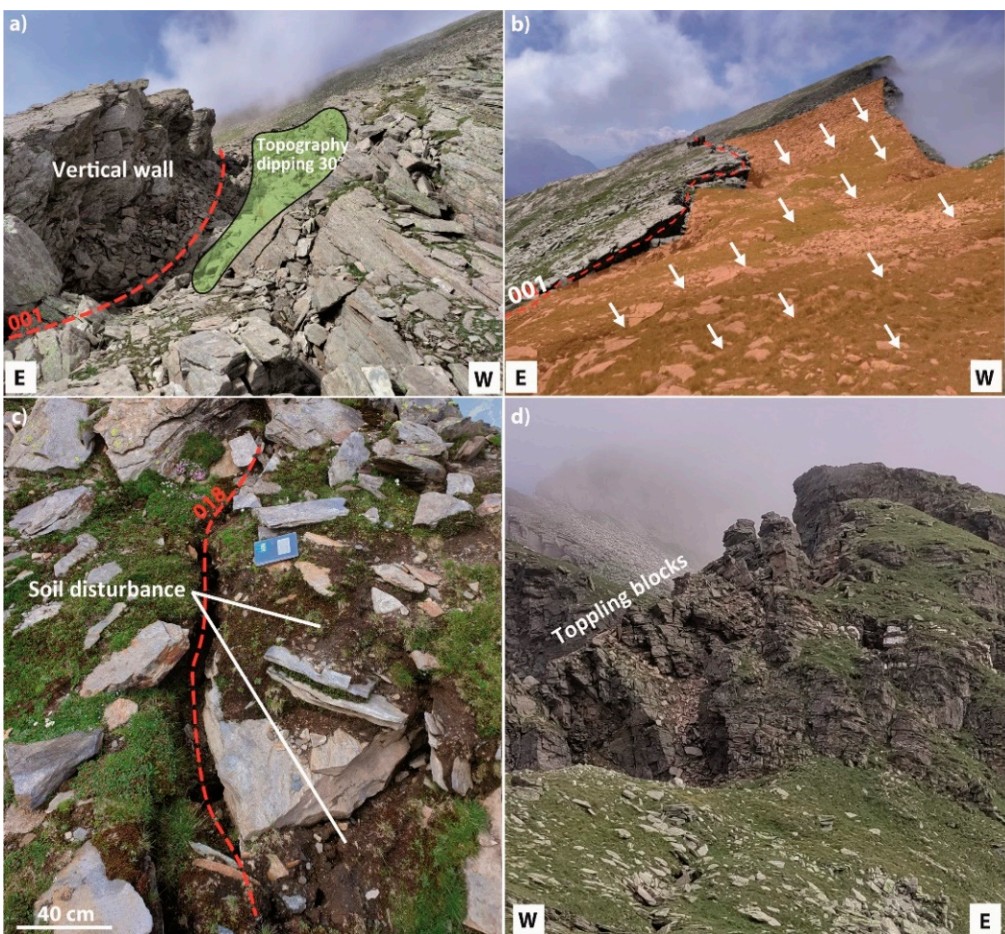

**Figure 3.** Photos depicting the main characteristics of the Cima del Simano instabilities. (**a**,**b**) Main open fracture; (**c**) Traces of movements in one of the openings near the crest; (**d**) Fractured blocks in toppling. Their location is shown in Figure 1c.

## 3. Materials and Methods

### 3.1. Material and Monitoring

Several different remote techniques were used to characterize the rock mass and monitor the various movements at different spatial scales (Table 1). The canton of Ticino started monitoring in 2006 using manual extensometers in open cracks to measure their aperture every year (Figure 1c,d). A more intensive monitoring started in 2021 with LiDAR and GB-InSAR campaigns and the processing of the Sentinel-1 satellite radar images by Gamma AG. Figure 4 presents the temporal coverage for each remote sensing technique used in the study.

**Table 1.** Remote sensing techniques used in the study for monitoring.

| Remote Sensing Technique | Acquisition Frequency | Spatial Resolution | Minimum Measurable Displacement | Range Acquisition Distance | Device Used | Movement Types Recorded |
|---|---|---|---|---|---|---|
| Extensometers | 1/year | One-off measure | 1 mm | In fracture Daily monitoring from office | Measuring tape | Slow movements of big volume and extent |
| LiDAR | 1/month during 3 months in 2021 | 30 cm [1] | 30 cm | From valley 3.5 km | Riegl VZ6000 | Rockfall and toppling activity |
| GB-InSAR | 2/year in 2022, 3 consecutive months in 2021 | 3 m | ~1 mm | From valley 3.5 km Daily monitoring from office | Ellegis LisaLab with a 3 m synthetic antenna aperture | Slow movements of big volume and extent Superficial topplings |
| Satellite InSAR | 2 weeks | 5 m | ~1–2 mm | From orbit 693 km altitude | Satellite Sentinel-1 | Slow movements of big volume and extent |

[1] For LiDAR, corresponds to the point spacing.

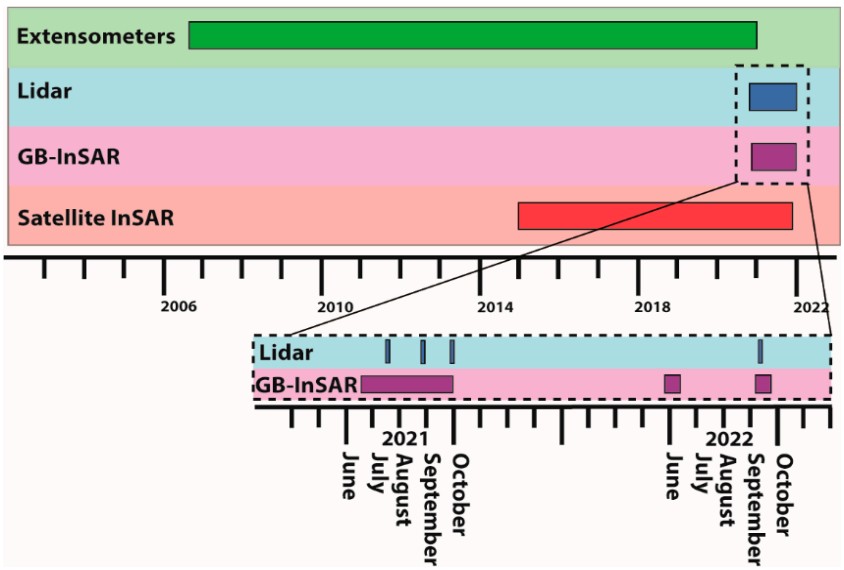

**Figure 4.** Temporal coverage of the field monitoring started in 2006 with extensometers measurements and more intensive monitoring since 2021 with LiDAR and GB-InSAR campaigns.

### 3.1.1. Drone Photogrammetry

From the summit, a DJI Mavic2 drone took pictures following nadir mission planning. The 700 images acquired were processed with Agisoft Metashape to get a 3D model, a Digital Elevation Model (DEM) and an orthophoto. A Ground Sampling Distance (GSD, [42]) of 3 cm was obtained. The orthophoto and DEM were used for the mapping of the fractures and depressions and the PC was used for the structural analysis of the main discontinuities.

### 3.1.2. Terrestrial LiDAR and High-Precision Panorama

A Riegl VZ6000 was used for the LiDAR acquisitions. In 2021, three acquisition surveys were performed at three different periods (July, September and October 2021). Due to poor weather conditions, only one acquisition was performed the year after, in September 2022. For each acquisition campaign, the LiDAR was installed at three different locations (Figure 1b) to reduce the number of hidden zones (occlusions, [43]). For all the acquisitions, the position of the LiDAR at 3.5 km from the summit of Cima del Simano, the Laser Pulse Repetition Rate (PRR, 30 kHz), the vertical and horizontal angle step widths (0.005°), the resolution (~30 cm for the upper part) and the average number of points (~30 million points) were all kept constant.

The LiDAR PCs were imported in Cloud Compare and aligned using the Iterative Closest Point-based (ICP, [29]) algorithm which minimizes the Root Mean Square Error (RMSE). Their Point-to-Point nearest orthogonal distances were then computed using the Least Square Plane model and the 10 nearest neighbor points (kNN). Since the average spacing distance between points varies between 20 cm and 30 cm, only points with a distance between the points of successive PCs greater than 20 cm were extracted [21] and only clusters of at least 5 points were considered as moving or fallen blocks and analyzed.

The PCs between July 2021 and September 2021 were first compared. The detected movements correspond to rockfalls or rock topplings. To ratify the detection of those moving blocks, the same comparison was made for the PCs between July 2021 and October 2021. If a moving block was detected by the first distance comparison as well as the second one, it confirms that the displacement corresponds to perennial rock movements and not to noise. In the last step, the distance computation between PCs of July 2021 and September 2022 was done to detect the fallen blocks and the debris accumulations over an entire year.

The LiDAR acquisitions were combined with the acquisition of high-resolution photos to produce panoramas using a full-frame Canon 5D camera with a 400 mm focal mounted on a tripod motorized base PiXplorer Clauss device [44], the latter assembling all the pictures using the Autopano Giga 3.5 software to produce the panoramas.

### 3.1.3. InSAR

The differential radar interferometry allows the detection of small and slow movements along the Line Of Sight (LOS, [3,45]), from the measurement of the phase difference of two back-scattered radar signals at two different times [46–48]. Several acquisitions permit obtaining the displacement time series [49–51].

#### Satellite InSAR

The Sentinel-1 constellation comprises two polar-orbiting satellites for a 1-week return period over the study area. The satellites operate in C-band (5.405 GHz), emitting a pulsed signal for a resolution of 5 m × 5 m. The Sentinel-1 radar images from 2015 to 2021 were analyzed by Gamma AG with the Interferometric Point Target Analysis (IPTA) software module [52] using the Persistent Scatterer Interferometry technique (PSI, [48,50]). Due to the slope orientation and the direction of the expected movements, satellite images from the descending orbit were used. Due to the presence of snow on top of the mountain from the late Autumn to the early Spring of the next year, only the images acquired from June to October were used and the images with poor co-registration with the chosen reference image were also retrieved from the analysis. The image on 17.10.2015 was used as a reference image for processing. The yearly speed displacements of the Persistent Scatterers (PSs) were calculated and interpolated with the Gamma software to obtain a speed displacement map for the satellite InSAR.

#### GB-InSAR

A LisaLab GB-InSAR was deployed at the foot of the Cima del Simano, in the village of Acquarossa (Figure 1b). This device is composed of a rail synthetizing a 3 m-antenna aperture. The measuring head moving along the rail emits a continuous radar signal in the Ku-Band (circa. 17 GHz) creating a radar image with a 3 min return period. The frequency bandwidth (BW, [53]) is 75 MHz for a minimum and maximum acquisition range distance of, respectively, 3100 m and 4200 m and a minimum and maximum acquisition azimuth distance of, respectively, 1300 m and 700 m. The measurements are stacked over 24 h to obtain a daily displacement time series with a better Signal to Noise ratio and to get rid of potential daily variations due to atmospheric effects [54,55]. To improve the atmospheric effects corrections and to proceed to the phase unwrapping, a zone along the radar range direction and assumed to be stable is selected.

In 2021, the GB-InSAR campaign lasted from mid-June to October. The atmospheric conditions near the summit were often very bad and the displacement time series ob-

tained were very noisy and, ultimately, only interferograms between the selected dates were analyzed, instead of the complete time series. For this reason, in 2022, it has been decided to proceed with two GB-InSAR campaigns of only two weeks each, one in June and one in September to avoid acquisitions in August, when frequent storms happen. Phase interferograms between the chosen dates in 2021 and 2022 were computed to detect displacements.

Since InSAR monitoring collects data only in the LOS of the device, GB-InSAR and satellite InSAR provide information along the line making an angle with the vertical of, respectively, 67° and 23° (Figure 5). According to the topography and the fracture orientations, the displacement direction of the unstable volume is expected to be down- and westwards. Thus, it is expected to find negative movement in the LOS of the satellite and positive ones in the LOS of the GB-InSAR.

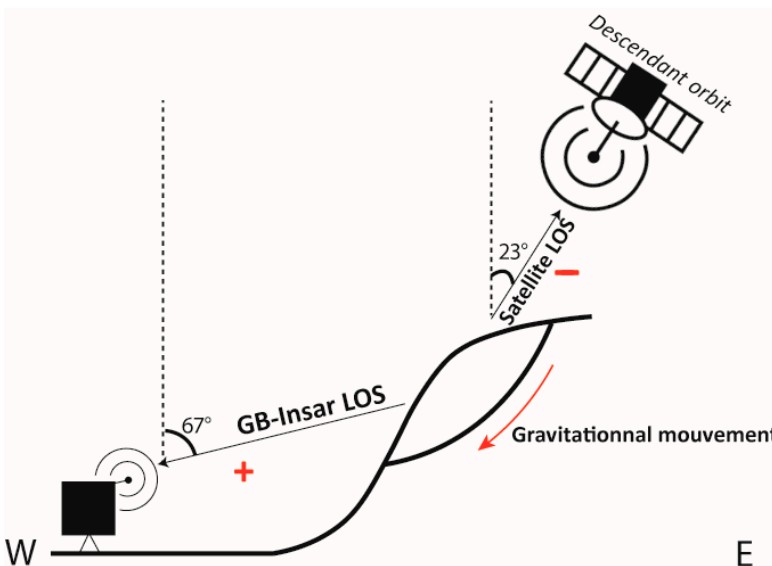

**Figure 5.** Expected displacement signs along the satellite and GB-InSAR LOS. GB- and satellite InSAR, respectively, provide data on sub-horizontal and sub-vertical displacements.

### 3.1.4. Temperature Monitoring

The variations of temperature were measured to estimate the presence of permafrost near the summit and the influence that could have the freezing–thaw cycles on the superficial instability destabilization. The air temperatures from two national meteorological stations, one located near the summit (SLFI, 2580 m) and one 130 m below (SFIM, 2450 m), were collected from the website of the Federal Office of Meteorology and Climatology of Switzerland (IDAweb, MeteoSwiss). In addition, thermologgers were installed in some fractures (Figure 1c). For each of the five locations selected, one logger was installed near the surface and a second in the fracture, at a depth of about 3 m.

### 3.2. Structural Analysis and Rupture Scenarios

A structural analysis of the main discontinuity sets' orientation (schistosity, faults, opened and brittle fractures, joint sets [56–59]) helps us to understand the mechanisms affecting the potential failures [6,60]. The aerial LiDAR DEM of the region (50 cm resolution) was collected from the website of the Swiss Federal Office of Topography (SwissAlti3D, Swisstopo, Wabern, Switzerland). The latter, as well as the LiDAR and drone SfM PCs, were analyzed with the software Coltop 1.8.9 to identify the main gneiss schistosity and joint sets orientations using a classical structural approach [21,56].

The results were confronted with the information from the geological map and field mapping. A kinematic analysis was conducted with Markland's tests [61] to highlight the discontinuity sets playing a key role in the destabilization and the movements observed at the summit of Cima del Simano. The MATLAB script developed by [62] was used for

the tests, locating, in the DEM, the areas susceptible to planar sliding, wedge sliding and toppling after the discontinuity sets found with Coltop. A low friction angle of 15° was used to account for the potential pore pressure and the filling of the fractures with fine materials. For the tests of planar sliding and toppling, lateral limits are set to 20°.

Then, scenarios on possible instability extents were outlined based on the topography and the monitoring results and drawn on ArcGIS Scene [63] before applying the Slope Local Base Level algorithm (SLBL, [64,65]). This technique is aimed at estimating the volume and thickness of a delimited unstable volume, giving a curvature tolerance C to the instability base level. C is calculated with the equation $4\frac{e}{Lrh}\Delta x^2$, with $e$ being the thickness, *Lrh* the length of the surface of the rupture and $\Delta x$ the resolution of the DEM.

## 4. Results

### 4.1. Monitoring Results

#### 4.1.1. Extensometers Results

The extensometer measures (one measure per year) provide good indications of the crack's activity. Figure 6 displays the evolution with time of the aperture of the cracks, the ones marked with a star are located in the major crack '001' and the others in the fractures are situated close to the crest. The crack '001' is inactive since the movements are insignificant. However, the aperture of the cracks near the crest increased, evidencing the instabilities whose mechanism, extent and volume are defined in this study.

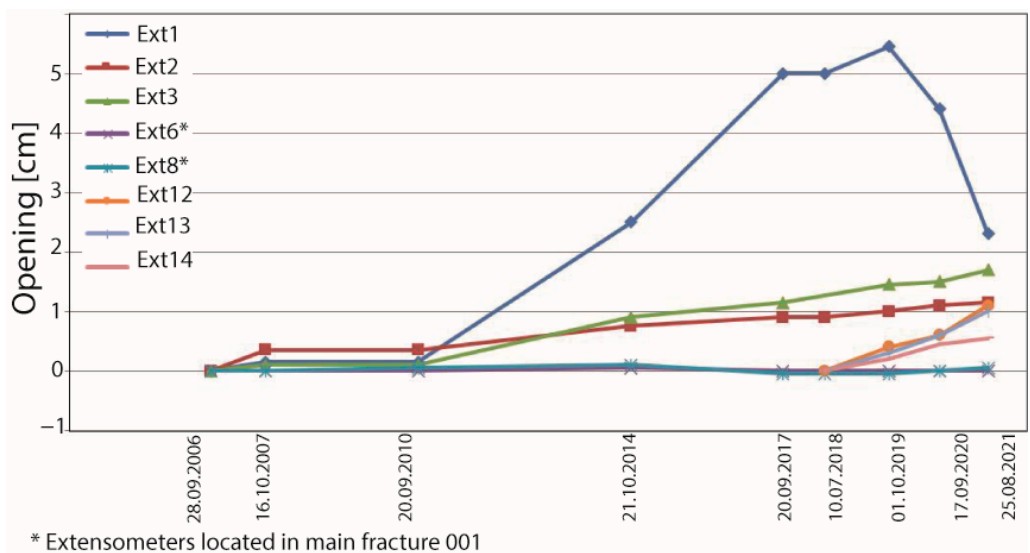

**Figure 6.** Fractures opening measurements since 2006, provided by the Ticino Canton. The main fracture 001 seems stable (corresponding to Ext6 and Ext8). The fractures closer to the crest seem active (Ext1, Ext2, Ext12, Ext13 and Ext14).

#### 4.1.2. LiDAR Results

The distance between the two PCs, acquired in July and September 2021, was computed and only points with a distance greater than 20 cm and gathered in clusters were extracted, highlighting zones where significant rockfalls or movements occurred in the steep cliffs of Cima del Simano (Figure 7a). With a detailed analysis of the zones of interest, it was then possible to discriminate the blocks that fell (the points forming the volume are no longer present, Figure 7e) from those that were undergoing toppling or sliding (the points forming the volume are still apparent, but have undergone a small shift, Figure 7d). The highlighted zones found in 3D were then reported on a panorama of the cliff to better locate them (Figure 7c).

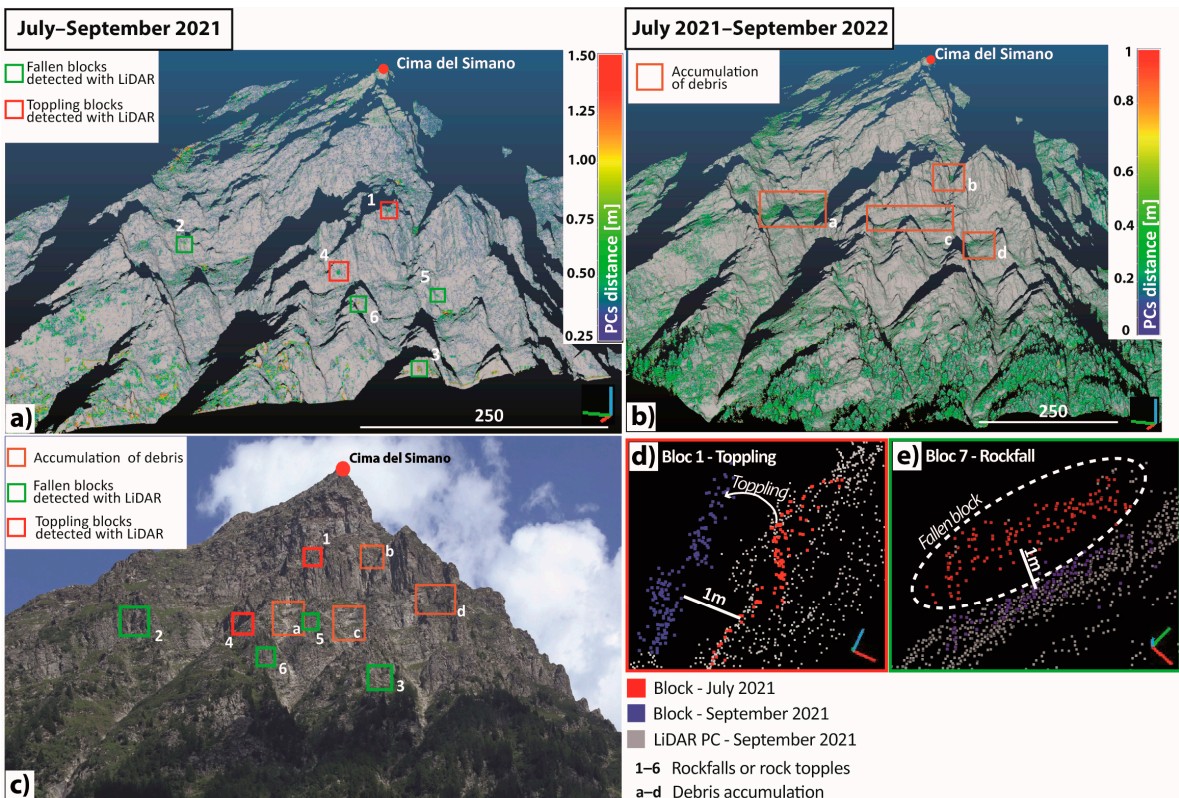

**Figure 7.** Rockfall sources, topplings and accumulation of debris detected by LiDAR PCs comparison. (**a**) Distance between PCs acquired in July 2021 and September 2021 and detection of rockfalls and toppling blocks. (**b**) Distance between PCs acquired in July 2021 and September 2022 and detection of debris from rockfalls. (**c**) Rockfalls, topplings and debris accumulations reported on a panorama. (**d**) Details of block 1 in toppling: the points forming the block in September (blue) are farther from the cliff than those forming the bloc in July (red). (**e**) Details of block 7 that fell between July and September 2021: the points belonging to the PC of September (blue) are aligned with the vertical cliff and thus belong to the latter, whereas the points from the PC of July (red) delineate the fallen block.

The same analysis was applied to PCs acquired between July 2021 and September 2022. This time, in addition to some rockfalls and toppled blocks that could be distinguished, wider areas where debris accumulated from small rockfalls, as shown in Figure 7b, were reported on the panorama (Figure 7c).

### 4.1.3. GB-InSAR Results and InSAR Interpretations

A two-month interval interferogram was created (Figure 8a,b) between June 2021 and August 2021 from the GB-InSAR acquisitions. The time difference was too short to detect movements of a potentially deep landslide. However, it highlights two blocks toppling, the top of the moving blocks showing more displacements than the bottom (Figure 8c,d). Those toppling movements are too small (between 4 mm and 10 mm) to be detected on the LiDAR PCs comparison.

The 10-month interferogram computed for the dates between September 2021 and July 2022 was unwrapped to obtain a displacement map along the LOS, oriented at 67° toward the vertical and in the direction 281°. It shows up a surface of slow movements located below the summit, with displacements toward the GB-InSAR between 4 mm and 13 mm (Figure 9), corresponding to a slow block sliding.

The Sentinel-1 PSs analysis, over 6 years, shows the same ranges of displacement velocity, between 4 mm and 13 mm per year along the LOS (23° toward the vertical), which prolongates about 60 m behind the crest (Figure 10). The displacements measured with the two techniques are coherent and seem constant over the years.

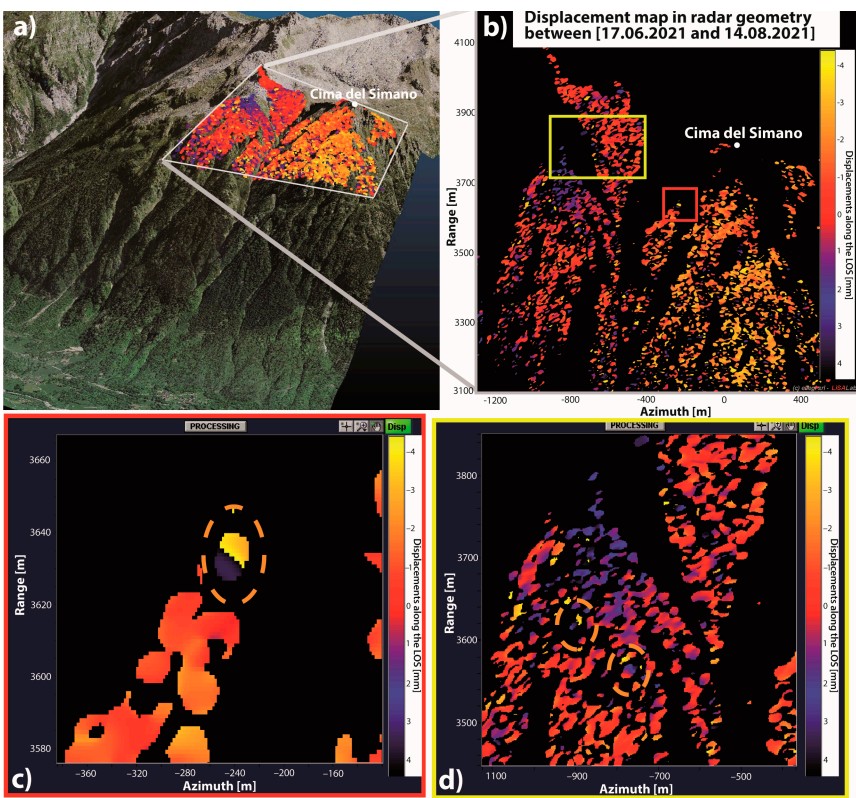

**Figure 8.** Wrapped GB-InSAR interferogram between June 2021 and August 2021 highlighting toppling blocks. The positive displacement values correspond to displacements toward the GB-InSAR. (**a**) The interferogram in geographical coordinates draped on a 3D orthophoto. (**b**) Interferogram in radar geometry (**c**). (**d**) Zoom on zones where toppling movements are detected. The upper part of each toppling block shows negative displacements due to wrapping effects.

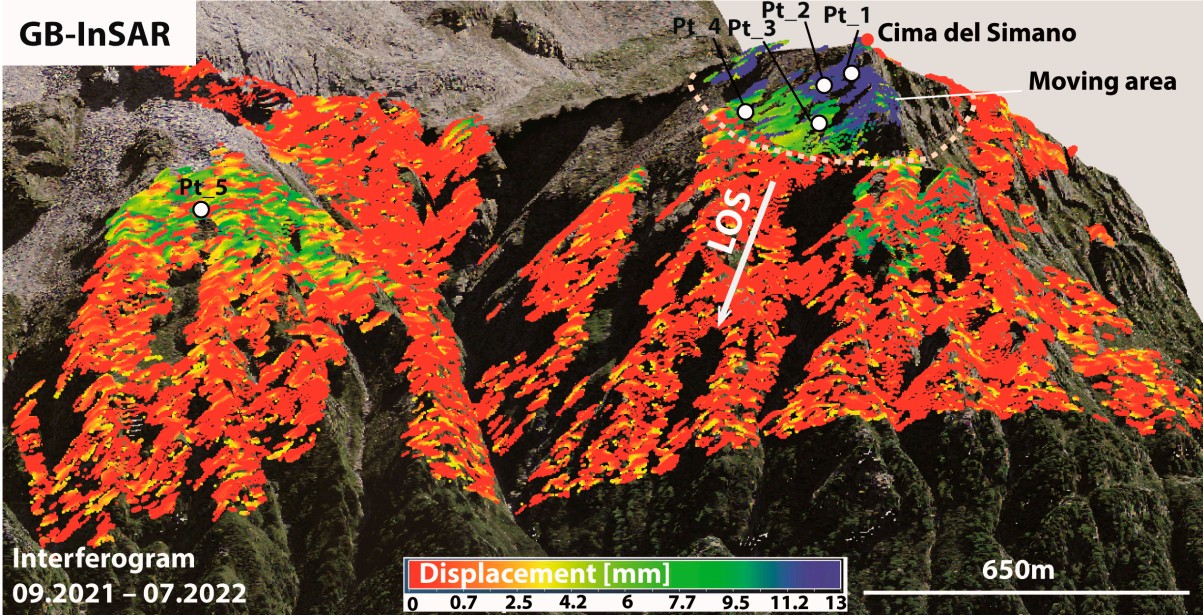

**Figure 9.** Unwrapped GB-InSAR interferogram between September 2021 and July 2022, highlighting an unstable zone with slow movements on top of the mountain. The displacement speeds of points Pt_1 to Pt_5 are plotted in Figure 13.

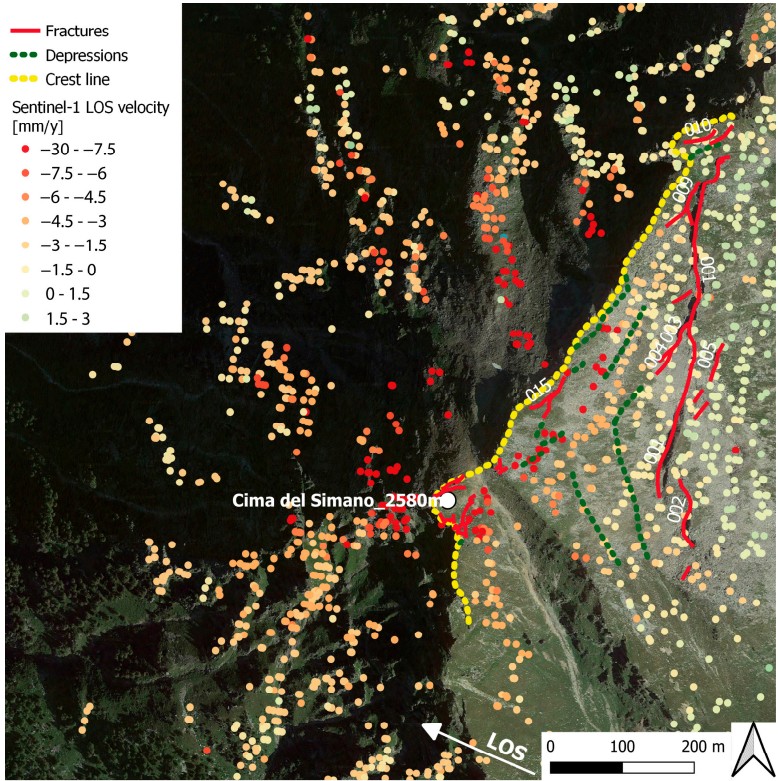

**Figure 10.** Sentinel-1 PSs speed displacement map over the study area, from the IPTA processing by Gamma AG of the radar images acquired between 2015 and 2021.

The GB- and satellite InSAR speed displacement maps were then merged in the same 3D map where the moving surface is easily identified (Figure 11). This encompasses an area of a wider extent; the triggered mechanism could be a deep-seated landslide.

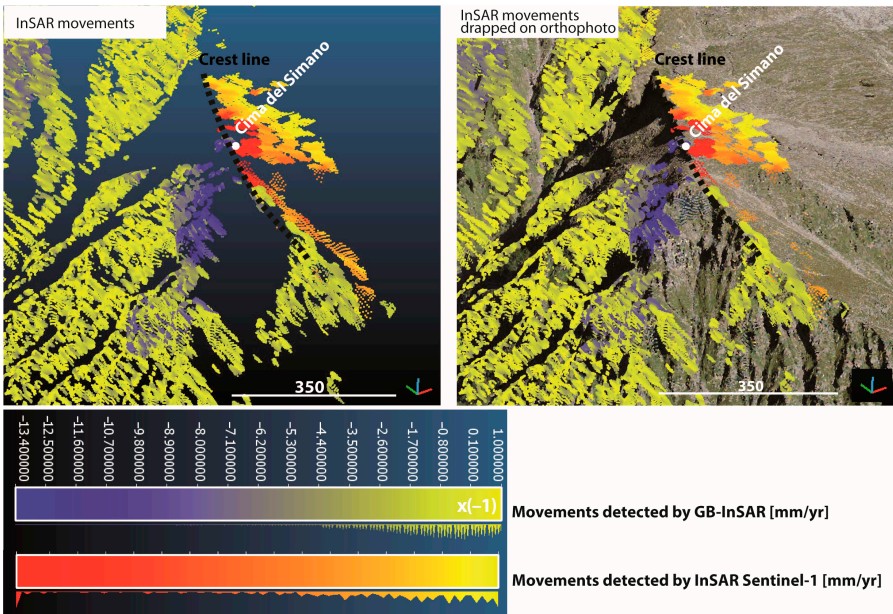

**Figure 11.** Spatial combination of the GB-InSAR (blue) and the satellite InSAR (red) displacements in the same 3D map delimiting an area with slow movements. The GB-InSAR movements must be multiplied by (−1); the movements are directed toward the GB-InSAR. GB- and satellite InSAR results are coherent, showing a similar range of displacement. The combination of the two results highlights the continuity of movements between the deep slope and the back slope.

### 4.1.4. Weather Results

After one year of acquisition with a frequency of one measure per two-hour, the thermologger data were gathered and the variations plotted in Figure 12 for the three months of June, July and August 2022, since it has been noticed that the loggers had difficulties registering the temperatures when the latter were below 0 °C for an extensive period of time. The temperatures recorded by thermologgers near the surface (Up) show an average variation between 8 °C at night and 15 °C during the day, while those recorded by the thermologgers deep in the fractures (Down) are more stable, around 6 °C throughout the day. The thermologgers $TH3_{down}$ registered negative temperatures in June.

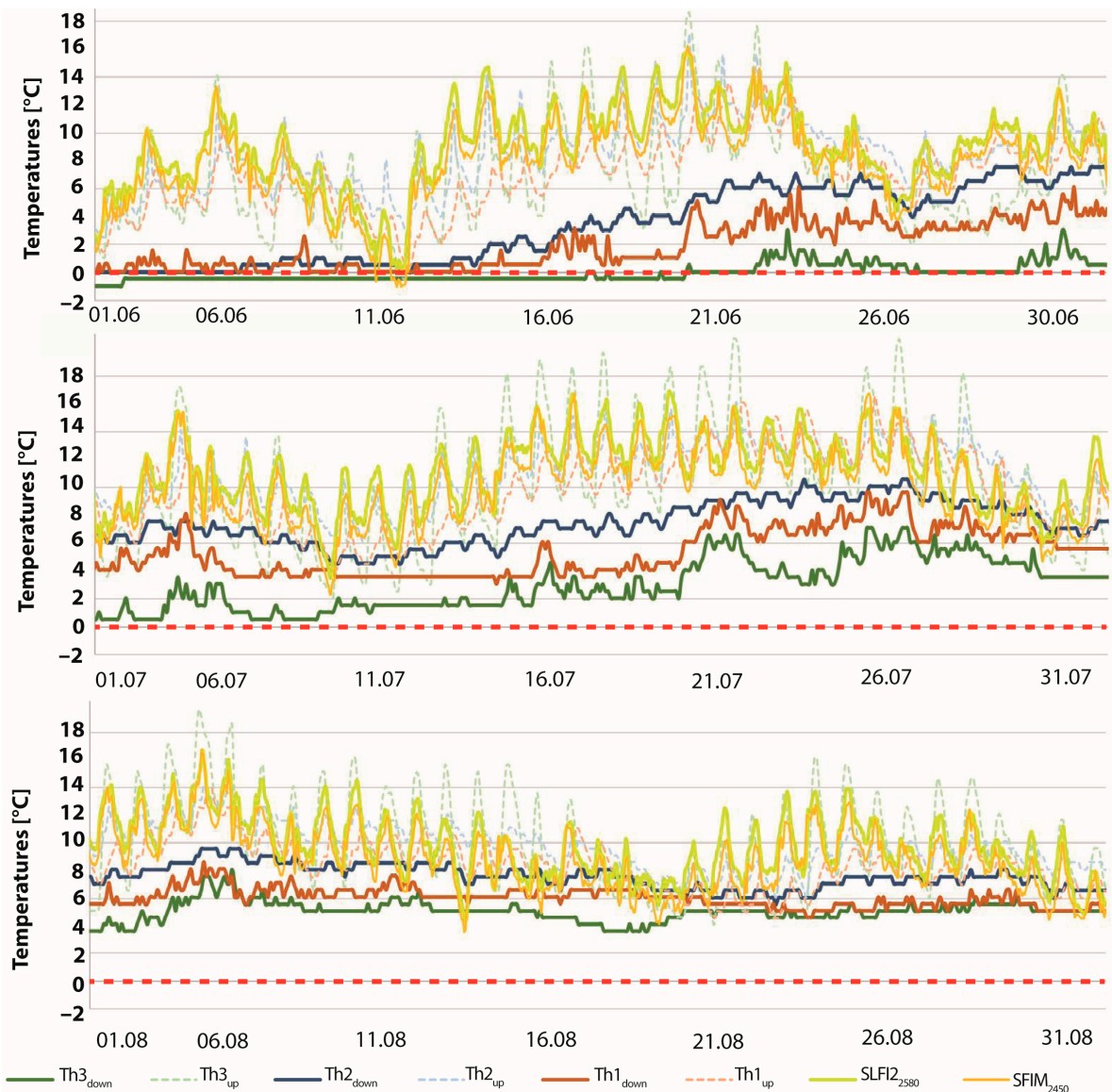

**Figure 12.** Variation of temperatures registered by the thermologgers (Figure 1c) and by the two national meteorological stations SLFI2 (2580 m) and SFIM (2450 m). Up for thermologgers close to the surface and Down for those deep in the fractures. The red dash line is the limit between negative and positive temperatures. The Down thermologgers register less variations of temperature, the latter remaining close to 0 °C.

Additionally, height points located on the cliff (Figure 9) and showing signs of movements according to the GB-InSAR were selected and their displacement speeds during

the 4 months of GB-InSAR monitoring in 2021 are plotted (Figure 13a), as well as the temperatures and precipitations recorded by the meteorological station SLFI2 (Figure 13b).

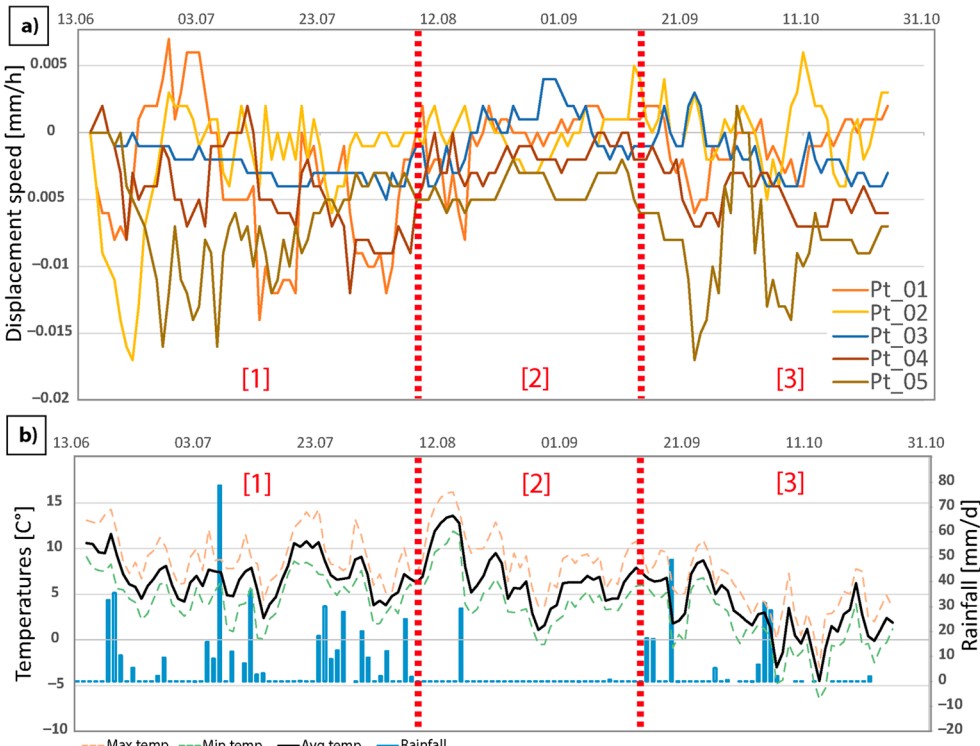

**Figure 13.** Correlation between GB-InSAR speed displacement and meteorological condition during the Summer 2021. (**a**) Variation of displacement speed for 5 points located on Figure 8. (**b**) Temperatures and precipitations measured by the national meteorological station. During the periods '1' and '3', the precipitations were more intense and the displacements more important than during period '2'.

*4.2. Structure Analysis*

With the PCs imported in Coltop, two main joint sets j1 and j2 were found (Table 2b). The j1 dip/dip direction measured from the LiDAR, drone SfM and Swisstopo DEM PCs are, respectively, 72°/180°, 70°/190° and 45°/173° and for j2, 65°/285° (LiDAR), 56°/283° (drone) and 46°/268° (Swisstopo DEM).

On the Swisstopo PC, an additional discontinuity set j3 is found, oriented at 47°/349°, and on the drone one, the joint set corresponding to the gneiss schistosity s1 is also detected with a dip/dip direction of 26°/080°.

In the field, 131 orientations of discontinuity were sampled and classified in three distinct sets corresponding to the two joint sets described above, j1 and j2, and to the schistosity s1. Thirty-eight measurements were not classified (Table 2a).

The two main vertical fracture lineaments depicted in Figure 1, named Lineament 1 and Lineament 2 in Table 2c, are also plotted in the stereonet. The geological map [41] highlights two faults, fault 1 (76°/246°, plotted in Figure 1) and fault 2 (59°/181°, located 100 m south of the summit), and the gneiss schistosity s1 (20°/035°).

The plotted orientations are dispersed over the entire stereonet. This stereonet pattern is characteristic of a rotational deep-seated landslide because the fracturing is high (Figure 3) and the discontinuities are spread in all directions (Figure 14c, [56,66,67]). Nevertheless, the sets j2 and j3 could also form a sliding wedge along the sliding line oriented at 51°/127° (Figure 14b).

**Table 2.** Main joint families plotted in the stereonet Figure 14a. (**a**) Manual measurements in the field. (**b**) Families extracted from the analysis of drone SfM, LiDAR and Swisstopo PCs with Coltop. (**c**) Information extracted from the orthophoto and the geological map.

| (a) Measurements in the field | | |
| --- | --- | --- |
| | Dip/Dip Direction | Quantity |
| j1 | 73°/178° | 25 |
| j2 | 89°/294° | 18 |
| s1 | 19°/018° | 50 |
| not classified | - | 38 |

| (b) From PCs analysis with Coltop | | | |
| --- | --- | --- | --- |
| | Dip/Dip Direction | | |
| | LiDAR | Drone | Swisstopo DEM |
| j1 | 72°/180° | 70°/190° | 45°/173° [2] |
| j2 | 65°/285° | 56°/283° | 46°/268° [2] |
| j3 | - | - | 47°/349° [2] |
| s1 | - | 26°/080° | - |
| Intersection j2/j3 [1] | 42°/197° | 46°/211° | 39°/234° |

| (c) From the geological map and orthophoto | | |
| --- | --- | --- |
| | Dip/Dip Direction | Orientation |
| s1 | 20°/035° | - |
| fault 1 | 76°/246° | - |
| fault 2 | 59°/181° | - |
| Lineament 1 | - | 85° |
| Lineament 2 | - | 330° |

[1] Plunge/Trend of j3 from Swisstopo DEM with j2. [2] Joint set orientations used in Markland's tests (Figure 15).

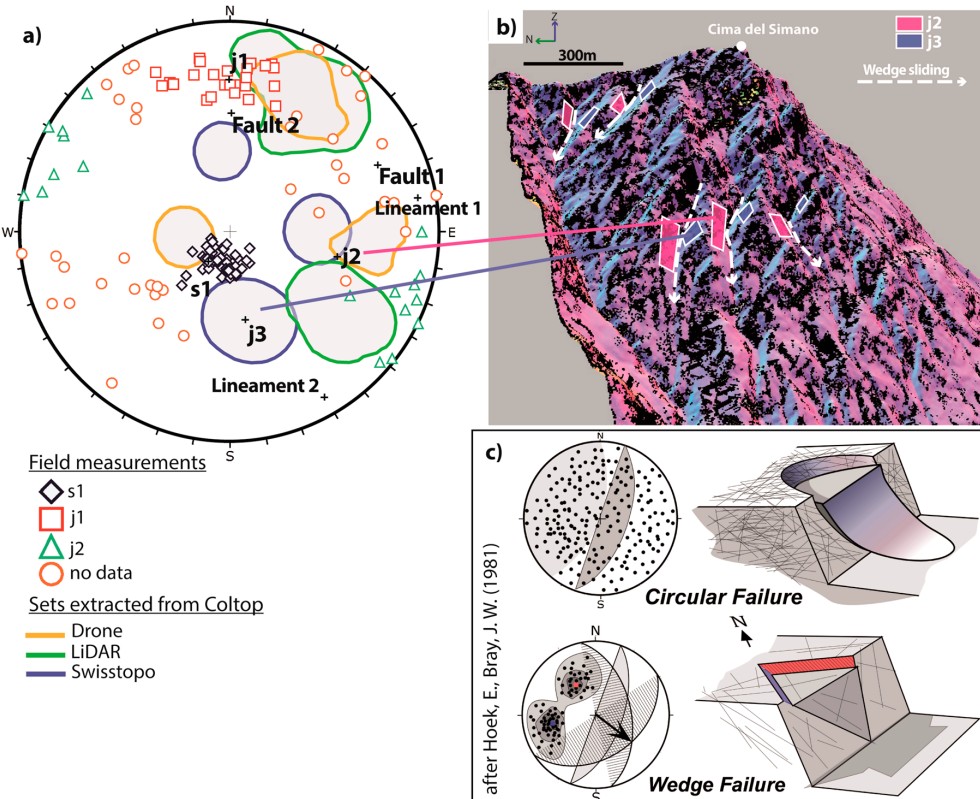

**Figure 14.** Structural analysis. (**a**) Lower hemisphere stereonet of the discontinuities measured on the field and on the Swisstopo DEM, the LiDAR and the drone PCs with Coltop. (**b**) Planes j2 and j3 forming a wedge located on the Swisstopo DEM PC. (**c**) Geometric condition and stereography pattern in the case of a rotational slope failure and of a wedge failure, after [56].

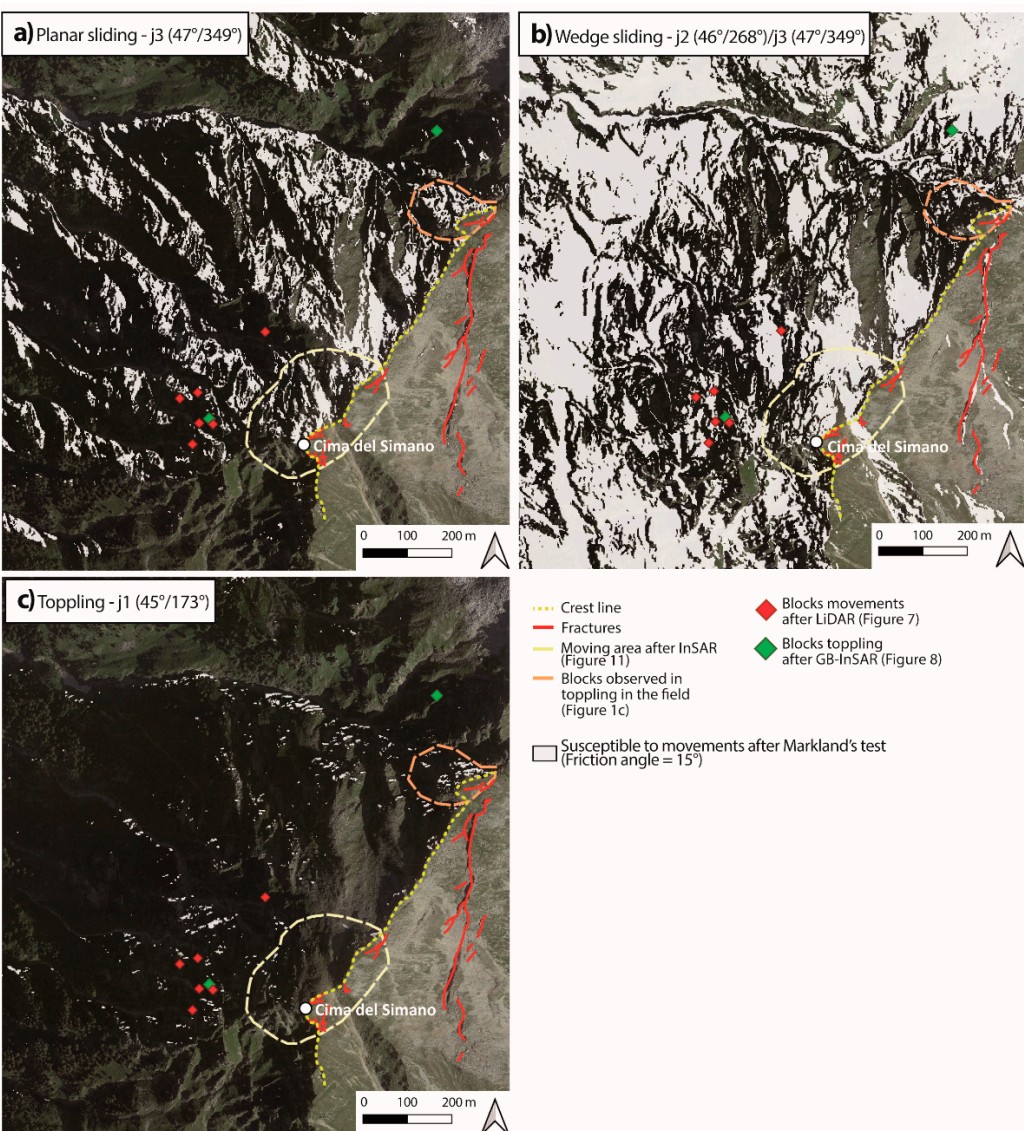

**Figure 15.** Markland's tests with the discontinuity sets detected in the Swisstopo DEM for a friction angle of 15°. (**a**) Susceptibility to planar sliding considering planes from j3. Angles of lateral limits are 20°. (**b**) Susceptibility to wedge sliding involving j2 (46°/268°) and j3 (47°/349°). (**c**) Susceptibility to toppling with planes from j1. Angles of lateral limits are 20°.

The Markland's tests were conducted with the three discontinuity sets extracted from the Swisstopo DEM. They confirm the susceptibility to planar sliding along planes belonging to joint sets j1, j2 (Supplementary Figure S1a,b) or j3 (Figure 15a) to wedge sliding involving j2 and j3 (Figure 15b) and highlight the locations of potential toppling blocks involving joint sets j1 (Figure 15c) and j3 (Supplementary Figure S1c).

The fallen and toppled blocks detected by LiDAR (Figure 7) and GB-InSAR (Figure 8) are located within areas prone to movement. The region exhibiting signs of displacement after InSAR analysis (Figure 11) is susceptible to both planar and wedge sliding. In the case of planar sliding, the primary joint set responsible for the destabilization is j3. The locations of the blocks prone to toppling in Figure 15c correspond to the positions of the blocks observed toppling in the field (Figure 1c), with j1 as the primary joint set controlling the movement.

## 5. Discussion

### 5.1. Weather Impact

#### 5.1.1. Temperatures Analysis Results

The temperatures recorded in the fractures by thermologgers located near the surface (Up) are correlated to the temperatures of the national station installed at the summit, the cross-correlation coefficients [68,69], ranging between 0.65 and 0.77 for 1600 measures during the summer. A cross correlation between the Up and Down temperatures at the same location is also performed, showing the maximum correlation for a time delay equal to zero, with values ranging from 0.61 (Th3) to 0.85 (Th2).

The negative temperatures registered by $TH3_{down}$ suggest the potential presence of permafrost at depth, as suggested by the Swiss Federal Office for the Environment (OFEN, [70]) on the Swisstopo portal (Figure 16). But with the climate changes it may have disappeared, causing the destabilization of the surface, and the recent acceleration of global warming could increase this destabilization.

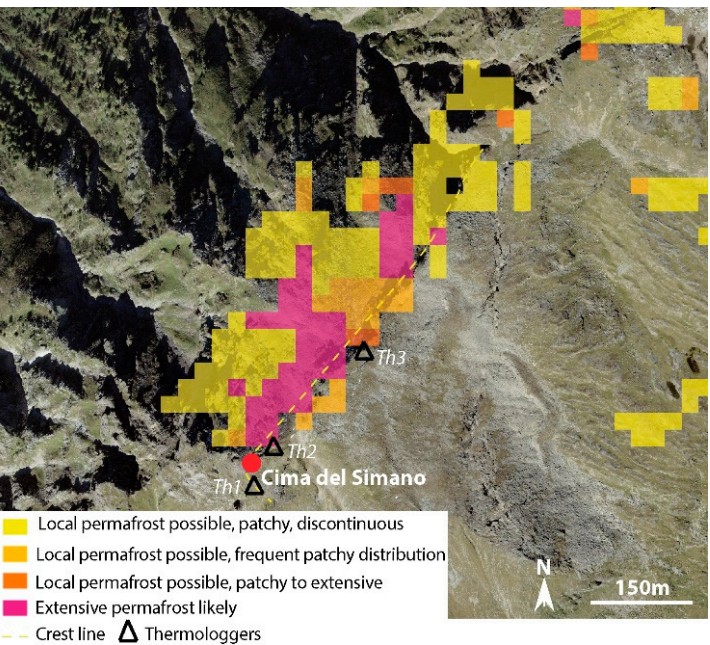

**Figure 16.** Permafrost map from [70] suggesting the presence of permafrost where the thermologgers were installed.

#### 5.1.2. Rainfalls Analysis Results

The data recorded by the GB-InSAR are very noisy. Nonetheless, it is possible to retrieve some trends. The months of June and September (periods '1' and '3' in Figure 13b) featured important rainfall events. Those periods are also marked by more intense displacements (Figure 13a). Conversely, the period '2' with fewer precipitations also shows smaller displacements. Rainfall events seem to be an important factor influencing the superficial sliding or toppling of blocks, but those results must be verified with additional measures to validate that the important detected movements are real displacements and not noise due to a radar signal dephasing occurring when the atmospheric effects are important, as could be the case during heavy rainfalls.

### 5.2. Limitations of the Manual Extensometers

The dates and times of measurements are not the same over the years. They do not get rid of potential variation of the openings due to rock dilation caused by thermal fluctuations, as suggested by [71]. To enhance the quality of the results and capture data reflecting daily variations, employing automated extensometers to record hourly measurements of fracture openings is a viable option. This approach would yield valuable additional insights into

site monitoring. When combined with weather stations and in situ thermologgers, it could help evaluate potential correlations between deformations and meteorological factors.

### 5.3. Scenarios for Rockfall and Rockslide Hazards

Eight scenarios of potentially unstable volumes that could be released are proposed in Table 3 and Figure 17.

From the observations in the field (Figure 3d) and the LiDAR results (Figure 7), the most likely events to occur are the toppling of small blocks located near the crest, also confirmed by Markland's tests (Figure 15c). But one can also consider scenarios of potential sliding movements of larger volumes involving the small fractures found near the crest ('015', '017', '018' in Figure 1) and the joint families j1, j2 and/or j3 (Figure 15a). We considered two such scenarios of Small Sliding (volume < $10^5$ m$^3$), denoted 'SS1' and 'SS2' (Figure 17a). The SLBL algorithm [46] was applied with the assumption that the sliding surface is almost planar. The unstable volumes computed were, respectively, $2.30 \times 10^5$ m$^3$ and $2.59 \times 10^5$ m$^3$ (Figure 18b,c).

The analysis of the InSAR images also reveals slow-movement areas affecting a larger extent. The clear delimitation between the moving and stable compartments is highlighted by the Sentinel-1 displacement map, which corresponds, in the field, with the presence of the important depression D1 (Figure 2). Five different scenarios (from 'S0' to 'S4', Table 3) were proposed whose delimitations follow the depression D1. 'S0' and 'S1' (Figure 19) delimit, respectively, the areas with displacements recorded by the GB-InSAR and the satellite InSAR. Scenarios 'S2' to 'S4' suggest wider unstable surfaces, encompassing the moving area and the lower boundaries following slope variations visible in the topography at different altitudes (2200 m, 2000 m and 1900 m), which could correspond to the instability limits.

The volumes of those scenarios computed with the SLBL vary from $3 \times 10^6$ m$^3$ to $5 \times 10^7$ m$^3$. An additional deep-seated scenario 'S5' (Figure 20a) is proposed considering a wedge sliding between two plans from the discontinuity sets j2 and j3. The geometry of the scenario was created in Cloud Compare using three planes, two of them belonging, respectively, to the two discontinuity sets and the third one to a back crack oriented at 85°/315° (Figure 20b). The volume was estimated by meshing the extracted PC corresponding to the unstable compartment with the Poisson Recon tool ([72], Figure 20c).

Scenarios 'SS1' and 'SS2' are independent and their failure would not influence the potential failure of the other scenarios, which could fail in a separate event. Conversely, if one of the larger scenarios were to occur, the smaller volume scenarios would cease to exist.

**Table 3.** Description of the 8 proposed scenarios.

| Instability Failure Mode | Scenario | Length [m] | Surface [m$^2$] | Volume [m$^3$] | Mean Thickness [m] | Curvature Tolerance C | Arguments in Favor of the Scenario |
|---|---|---|---|---|---|---|---|
| Superficial movement, Sliding or toppling | SS1 [1] | 50 | - | 230k | - | 0 | Fresh soil in some fractures Blocks toppling visible in the field Toppling blocks detected by LiDAR and InSAR |
| Superficial movement Sliding or toppling | SS2 [1] | 70 | - | 251k | - | 0 | |
| Deep-seated rotational movement | S0 | 200 | 63,000 | 3.8M | 20 | 0.33 | Delimitated by the displacements detected by GB-InSAR only |
| Deep-seated rotational movement | S1 [1] | 300 | 87,500 | 4.3M | 30 | 0.043 | Delimitated by the displacements detected by GB- and satellite InSAR |
| Deep-seated rotational movement | S2 | 400 | 178,000 | 16M | 70 | 0.039 | Encompasses the moving area detected by InSAR Follow the topography along the isoline at the altitude of 2200 m |
| Deep-seated rotational movement | S3 | 450 | 241,000 | 22M | 80 | 0.033 | Encompasses the moving area detected by InSAR Follow the topography along the isoline at the altitude of 2000 m |
| Deep-seated rotational movement | S4 | 600 | 451,000 | 51M | 110 | 0.024 | Encompasses the moving area detected by InSAR Follow the topography along the isoline at the altitude of 1900 m |
| Deep-seated sliding constrained by two plans | S5 [1] | 320 | 76,200 | 7.7M | - | - | Delimitated by the displacements detected by GB- and satellite InSAR Constrained by two plans from j2 and j3 |

[1] Further described in Figures 18–20.

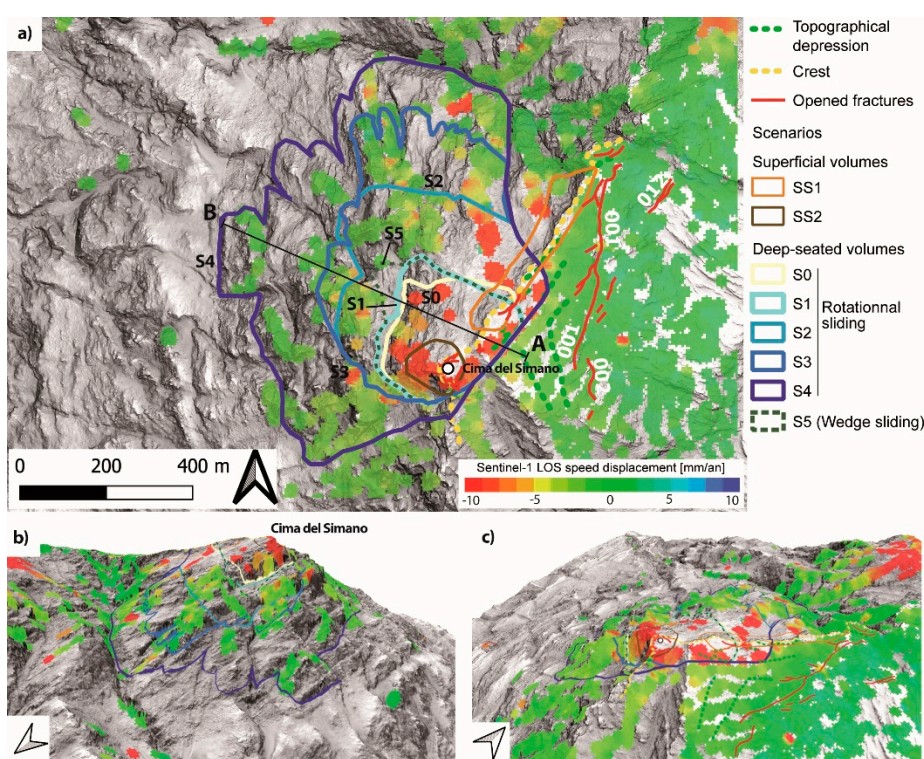

**Figure 17.** Delimitation of the eight scenarios. (**a**) In 2D. [AB] corresponds to the cross section of Figure 19d. (**b**), (**c**) In 3D.

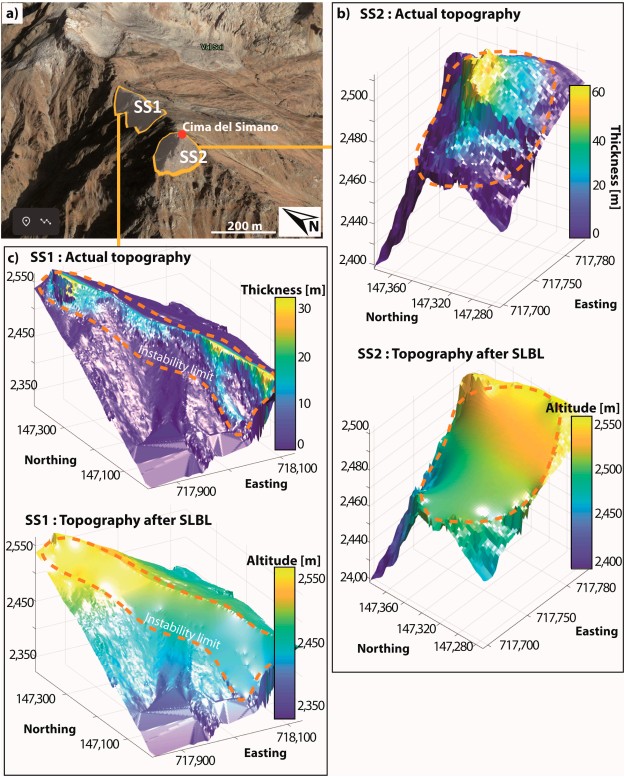

**Figure 18.** Rupture scenarios 'SS1' and 'SS2' for small volumes that could trigger rockfalls. (**a**) Potential instabilities outlined for the two scenarios. (**b**) and (**c**) SLBL results for scenarios 'SS1' and 'SS2' estimating, respectively, an unstable volume of $230 \times 10^3$ m$^3$ and $251 \times 10^3$ m$^3$ that could be released in the case of a slope failure.

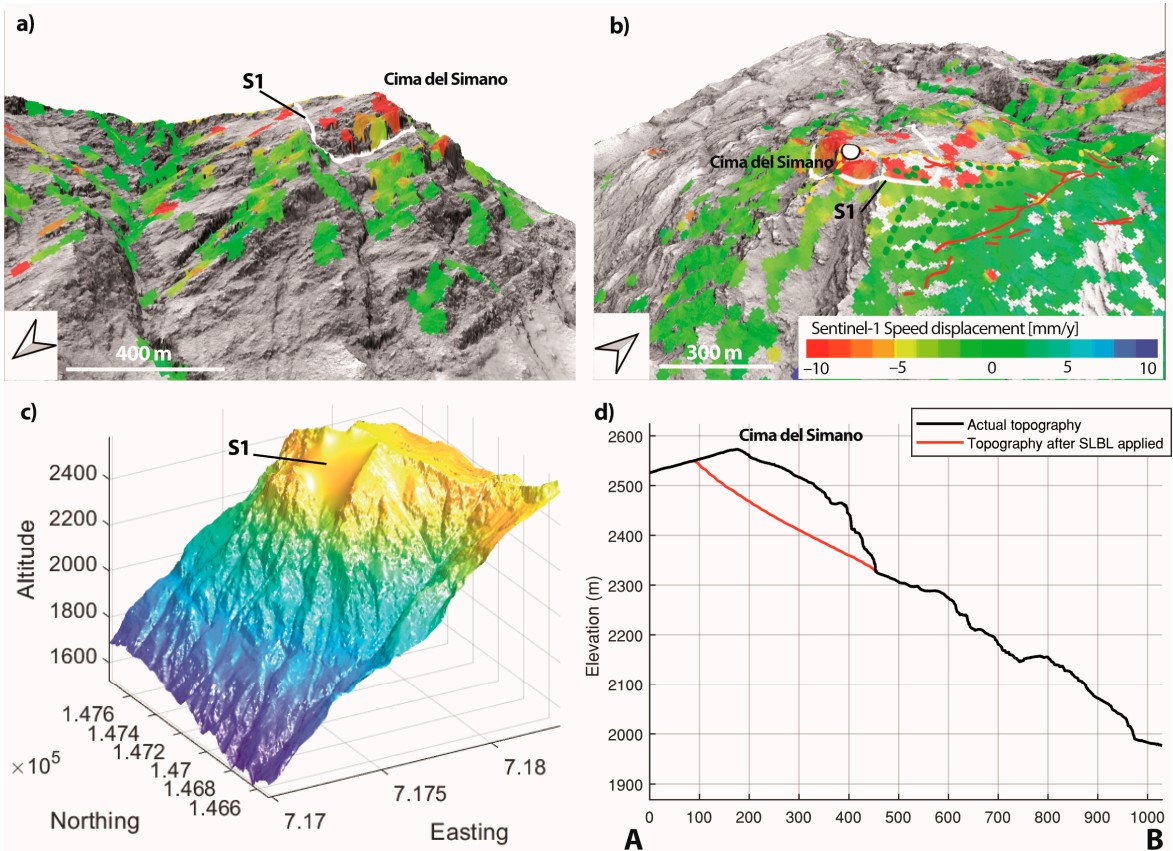

**Figure 19.** Detailed description of the rotational sliding scenario S1. (**a**,**b**) Delimitation drawn on 3D map corresponding to the moving area according to InSAR. (**c**) Topography after applying the SLBL algorithm. (**d**) Cross-section along the sliding direction line [AB] presented Figure 17.

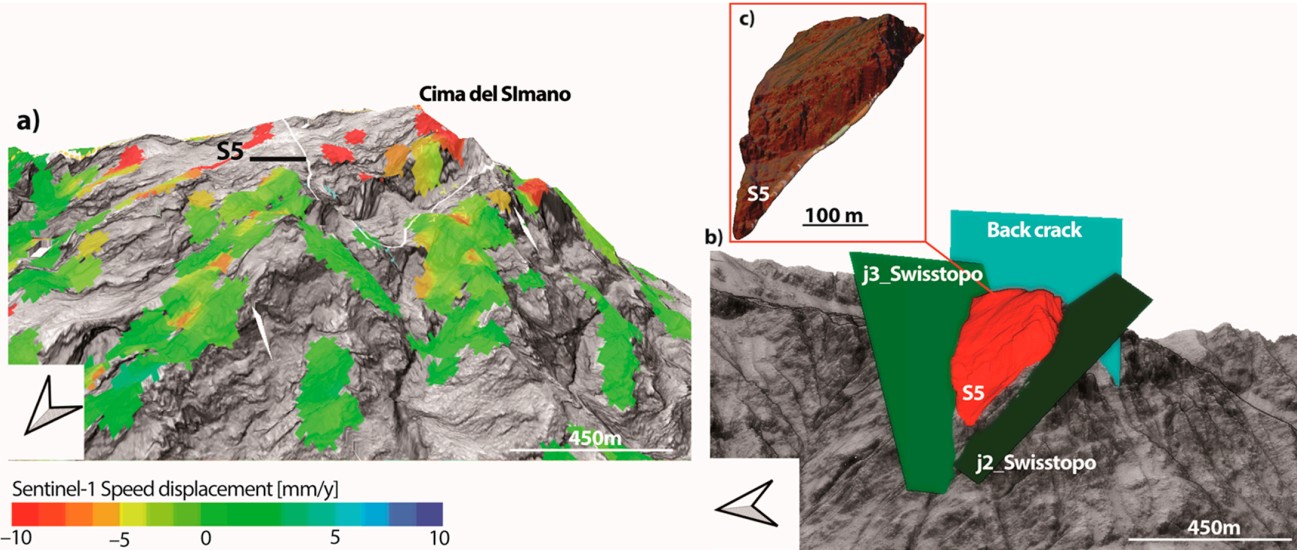

**Figure 20.** Detailed description of the wedge sliding scenario S5. (**a**) Delimitation drawn on 3D map corresponding to the moving area according to InSAR. (**b**) Geometry constraining the sliding of volume S5. (**c**) Extracted volume meshed with Cloud Compare.

### 5.4. Scenario SM3: Case of the Main Open Fracture '001'

The scenario involving the major open fracture '001' was named scenario 'SM3' and delimiting its volume assumes that the contour of the instability is defined by the fracture

'001' and encompasses a huge volume. In the field, several fractures were observed toward the North of the main fracture, and likely represent its prolongations. Figure 21 displays the limits suggested for the scenario SM3.

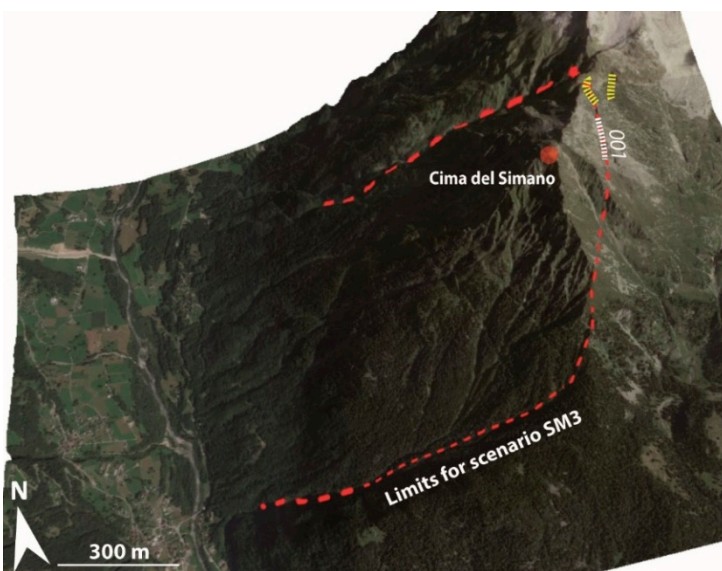

**Figure 21.** Hypothetical delimitation for the unstable compartment involving the main open fracture '001'. In white dash line, the location of the open fracture and, in yellow dash line, the prolongation of a fracture visible in the topography.

*5.5. Susceptibility Assessment*

With the proposed scenarios volume, extent and failure mechanism, in addition to the information on the deformation rates and the rockfall activity, an attempt can be made to locate the suggested landslide scenarios on a susceptibility chart, as in the one proposed by [73]. This chart gives a first estimate of the failure likelihood for each scenario, according to their deformation state and their deformation rate or rockfall activity (Table 4, Figure 22). This principle is based on the fact that different deformation types associated with different degrees of activity have different degrees of instability and susceptibilities to happen.

**Table 4.** Susceptibility for each scenario, based on different criteria.

| Scenario | Length [m] | Displacement Speed [mm/y] | Deformation Rate [%/y] | Deformation State (After [73]) | Susceptibility |
|---|---|---|---|---|---|
| SS1 | 50 | 9 | 0.018 | 2-2 | High |
| SS2 | 70 | 11 | 0.016 | 2-2 | High |
| S0 | 200 | 7 | 0.0035 | 2-1 | Moderate to high |
| S1 | 300 | 7 | 0.0023 | 2-1 | Moderate to high |
| S2 | 400 | 7 | 0.0018 | 2-1 | Low to moderate |
| S3 | 450 | 7 | 0.0016 | 2-1 | Low to moderate |
| S4 | 600 | 7 | 0.0012 | 2-1 | Low to moderate |
| S5 | 200 | 7 | 0.0028 | 2-1 | Moderate to high |

In this chart, the deformation state is defined by two values i–j, corresponding, respectively, to the relative deformations at the crest and at the toe of the instability; 0 for 'nearly insignificant', 1 for 'significant' and 2 for 'large'.

Scenarios 'SS1' and 'SS2' show clear signs of movement and rockfall activity for an average displacement speed, respectively, of 9 mm/y and 11 mm/y, equivalent to a deformation rate between 0.016%/yr and 0.018%/yr. The displacements are similar at the top and at the toe of the delimited scenarios. Observations in the field indicate that the rock

is highly fractured with signs of rockfall activity, which may be indicative of significant internal deformations. The deformation state is of type 2-2 and the susceptibility can be supposed to be high.

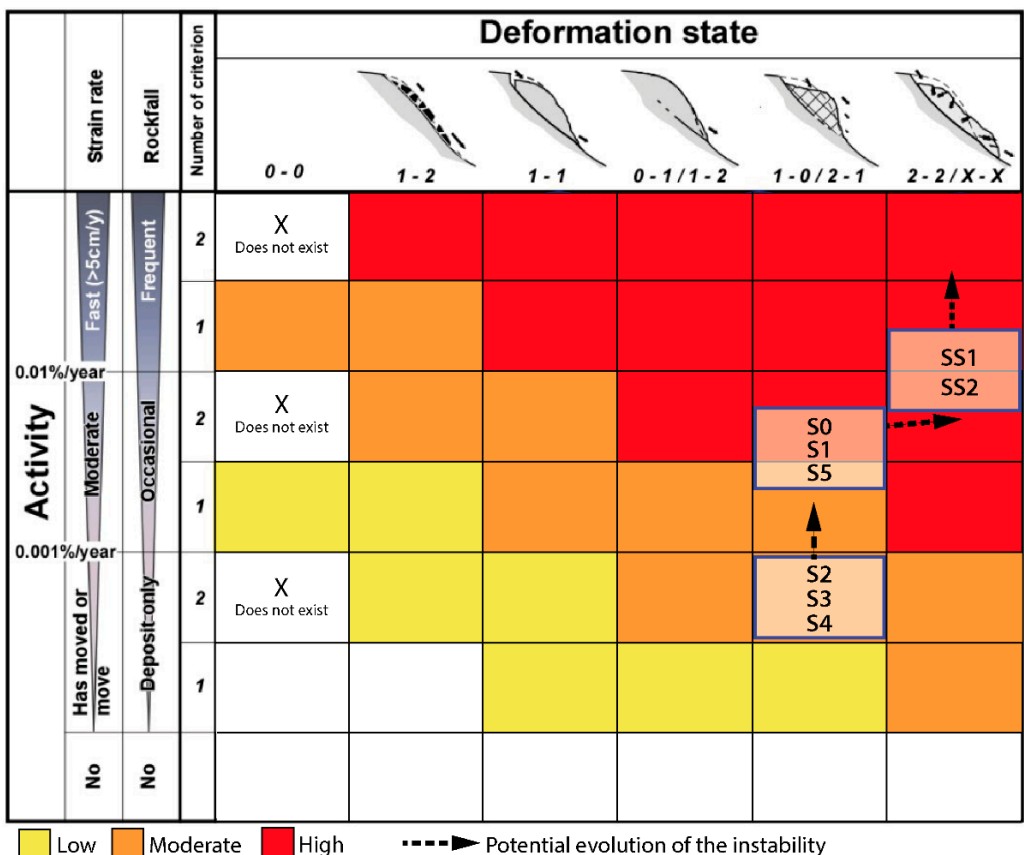

**Figure 22.** Attempt to locate Cima del Simano suggested instabilities on the susceptibility chart proposed by [73], based on the combination of information on deformation states and rates, and potential evolution of the instability.

For scenarios 'S0', 'S5' and 'S1', delimiting the moving areas, their total instability lengths are, respectively, about 200 m, 250 m and 300 m, for a displacement speed of 7 mm/yr in average, leading to a deformation rate of, respectively, 0.0035%/yr, 0.0028%/yr and 0.0023%/yr, with very little rockfall activity, mostly located at the crest. From the InSAR results, the instability experiences more deformation at its crest than at its toe. The deformation state could be 1-0 and the corresponding susceptibility is moderate to high.

Scenarios 'S2', 'S3' and 'S4', of greater length, do not show signs of movement at their toe, corresponding to a 1-0 deformation state. The average deformation rate is 7 mm/y, leading to a deformation rate, respectively, of 0.0018%/yr, 0.0016%/yr and 0.0012%/yr. The susceptibility is low to moderate.

With those considerations, the monitoring of this zone should be continued, by means of additional GB-InSAR campaigns or automated extensometers installed in some relevant cracks, to follow the potential evolution of the deformation rate and state. Scenarios 'SS1' and 'SS2' located at the crest are the most probable to happen and monitoring would help foresee the rupture by an increase of the deformation speed. Scenarios 'S0', 'S5' and 'S1''s deformation states could evolve to 2-2 if the toe shows more displacement in the future, increasing the susceptibility. Finally, the monitoring would help to verify if the scenarios of larger volumes, 'S2' to 'S4', become more probable to happen, by the propagation of the deformation to the toe. The potential impact of climate change on the weathering of the rock mass could also have an impact on the evolution of the deformation rate.



## 6. Conclusions

The monitoring strategy presented above highlights the value of combining several remote sensing techniques to describe an instability. Based on the IPTA analysis of the satellite SAR images, an initial estimate of the moving zones is obtained, which is confirmed by GB-InSAR, giving a similar order of magnitude of deformations. LiDAR monitoring assesses the rockfall activity, detecting the source areas and the toppling movements, which are not detectable by InSAR because of its resolution of 5 m. In any case, field mapping is necessary to detect clues (fractures, depressions, toppled blocks, and discontinuity orientations) regarding the instability mechanisms. From that information, combined with a structural analysis and the application of the SLBL method, potential failure scenarios can be defined in terms of volume and geometry.

This procedure was applied to the study of Cima del Simano and its instability hazards, giving relevant information on the different movement scales (from millimeter to centimeter) and types (superficial or in-depth). Two superficial instabilities, 'SS1' ($2.30 \times 10^5$ m$^3$) and 'SS2' ($2.59 \times 10^5$ m$^3$), showing signs of movements and soil disturbance can be considered as active and their volume could be released at once or by several small rockfall events. The accumulation of blocks due to this activity would probably be situated in the upper part of the cliff, but could trigger debris flow in future years.

Deep-seated instability scenarios, 'S0' to 'S5' ($3 \times 10^6$ m$^3$–$5 \times 10^7$ m$^3$), are also proposed. 'S0', 'S1' deep-seated rock instabilities and the wedge 'S5', delimited by the InSAR displacement results, correspond to a proven active zone with a mean displacement speed of 7 mm/yr. An attempt to qualify the scenarios suggests low to moderate hazard failure susceptibility. The weathering impact on the slope constraints and the erosions could be further investigated by means of numerical modeling, as already suggested by some studies [74–76]. Those investigations could help to assess the possible evolution of the instability in terms of hazard susceptibility.

**Supplementary Materials:** The following supporting information can be downloaded at: https: //www.mdpi.com/article/10.3390/rs15225396/s1, Figure S1: Markland's tests for planar sliding and toppling with the discontinuity sets detected in the Swisstopo DEM for a friction angle of 15°. Angles of lateral limits are 20°. (a) Susceptibility to planar sliding considering planes from j1. (b) Susceptibility to planar sliding considering planes from j2. (c) Susceptibility to toppling considering planes from j3. The SLBL routine used for the calculation of the volume for each scenario of rupture is available online in the GitHub public repository at: https://github.com/charlottewolff/ SLBL, accessed on 28 October 2023. An additional figure (Supplementary Figure S1) can be found in the online version of this article with the results of the others Markland's tests).

**Author Contributions:** Conceptualization, C.W., M.J. and A.P.; methodology, C.W., M.-H.D., M.J. and A.P.; validation, C.W., M.J. and M.-H.D.; formal analysis, C.W.; investigation, C.W., L.F., M.-H.D., C.R. and V.M.-S.; data curation, C.W. and C.R.; resources, M.J. and A.P.; funding acquisition, M.J. and A.P.; writing—original draft preparation, C.W.; writing—review and editing, M.J. and V.M.-S. visualization, C.W. All authors have read and agreed to the published version of the manuscript.

**Funding:** This research received no external funding.

**Data Availability Statement:** Data will be made available on request.

**Acknowledgments:** We thank Amalia Gutiérrez, Tiggi Choanji, Stefano Cardia and Valérie Baumann (Risk analysis group, University of Lausanne, Switzerland) for the GB-InSAR and LiDAR installations and acquisitions. We thank Roberto Gardenghi (New Celio Electronics GmbH, Dongio, Switzerland) for the automated extensometers installation. We thank Christophe Magnard and Tazio Strozzi (Gamma Remote Sensing, Gümligen, Switzerland) for helping in the processing and analysis of the satellite InSAR data. We thank Ticino Canton who mandated this monitoring study and provided the extensometer measurements and satellite InSAR data. We thank the commune of Acquarossa for helping for the GB-InSAR installation.

**Conflicts of Interest:** The authors declare that they have no known competing financial interest or personal relationships that could have appeared to influence the work reported in this paper.

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
