# Peer review of "Assessing the Hazard of Deep-Seated Rock Slope Instability through the Description of Potential Failure Scenarios, Cross-Validated Using Several Remote Sensing and Monitoring Techniques"

_remotesensing, doi:10.3390/rs15225396_

Round 1

Reviewer 1 Report

Comments and Suggestions for Authors

Dear authors, 

Congratulations for the paper, it was a real pleasure to read and evaluate it! It showed me a clear vision of the work, a detailed knowledge of the issue and a strong expertise in data and devices handling. As geomorphologist, I simply cannot find an alternative way to increase its value.   

Author Response

Dear reviewer, 

Thanks for your very enthusiastic answer. 

Please find attach the responses to the reviewer comments

Reviewer 2 Report

Comments and Suggestions for Authors

The reviewed manuscript describe the investigation of an unstable slope using a combination of field and remote sensing methods for characterization and monitoring, including LiDAR, InSAR, and photogrammetry. Based on monitoring data and the geomorphic characterization of the slope (namely, the Cima del Sanaro peak in the Ticinese Alps), multiple scenarios are introduced, which are then examined and their likelihood of occurrence inferred.

The manuscript is well presented, structured, and written (though some typos were noted that have been highlighted later in this review). I believe moderate revisions are needed to make the manuscript acceptable for publication. This is due to two main concerns I have with the data analysis, which I believe, when addressed, should make the manuscript more robust.

The first point is related to the fact that a preliminary stability analysis is missing. The scope of the paper is probably not to provide the reader with a detailed investigation of the processes underlying the stability, however, a simple kinematic analysis should be still be provided to a) highlight what are the discontinuity (sets) that play a role in controlling the instability observed in the field. For instance, the block toppling near the crest and, especially, the wedge sliding described for scenario 5. Presently, the paper presents some reference to potential failure mechanisms, but the easiest (if not the only) way to support the interpretation is through a kinematic analysis (i.e., Markland’s test). This should ideally be included in section 4.2 (and in figure 14, see line-by-line comment below).

The second point is related to the change detection analysis presented in section 4.1.2 (see also comment below). It appears that a simple cloud-to-cloud distance analysis was conducted, as the results shows only positive values. However, a M3C2 analysis would probably more suited in this case (the tool is also included by default in CloudCompare), as it would allow positive or negative changes to be highlighted (i.e., material loss and gain, as well as block displacement). If there is a reason for which such analysis was not undertaken (or described), this should be provided, otherwise this could be added (or better, it could substitute the current change detection analysis). With all probability, results will not change, but the readability of the figure 7 (as well as the ease of interpretation) would improve significantly.

The last point is related to the title, in which the authors refer to the assessment of “imbricated” deep seated rock slope instabilities. However, across the manuscript there is not mention of what it is meant by “imbricated” instabilities, at least in this context. It sounds like authors refer to “nested” instabilities, but does not appear to be the case in this manuscript, as the different instability scenarios are discussed as alternatives, rather than combining parts (an interesting line of enquiry would be “how do nested instability scenarios could appear in the susceptibility chart?”). Authors should clarify and expand the discussion of this aspect, or modify the title.

In the following, I also list some minor, line-by-line comments

Line 2: “rock slope” in title

Line 9: Check affiliation n. 3, there might be a typo

Line 37: “analyze/analyse”

Line 68: check “commune”

Line 76: please specify which figure

Line 79: please be more specific, by providing an average dip angle, instead of referring to “more” or “less steep”

Line 85 (figure): in 2c, it is difficult to understand where topographical depressions occur. This could also be related to the line color, as D1 seems demarked by a white, dotted line, which might be the outline of the green, dashed line shown in legend.

Line 88: Please provide detail about the basemap in tab 1c, e.g., Satellite imagery or UAV?

Line 109: in table, the LiDAR device might be a “VZ6000”?

Line 119: as above

Line 129: The point-to-point (P2P) distance provides the absolute distance between the point sets, regardless of the position of each PC. Indeed, figure 7 shows only positive distances between the July-September 2021 PCs and the July 2021-September 2022 PCs. What was the rationale for using a simple P2P change detection, versus, for instance, an M3C2 analysis (also available in CloudCompare) that can provide a positive or negative change (thus allowing an easier interpretation of the change detection analysis, distinguishing between volume loss, gain, and toppling where gain is observed in steep areas)? If better/clearer results are obtained, it could be worth modifying this paragraph and figure 7 – clarity will improve significantly, while results and interpretation will most likely not change.

Line 149: I suggest using “two different times” rather than “dates”, as this small introduction seems to refer to InSAR in general, not only satellite.

Line 218: perhaps “evidencing the instabilities”, rather than “witnessing instabilities”

Line 263: It is not very clear how the two datasets were merged. Was there any pre-processing to match the unit of displacement rates and/or to create a coherent map from two datasets with different LoS?

Line 280: in fig 11, the scalebar for the movement detected by the GB-InSAR should be in mm/y as well?

Line 290: perhaps “the loggers had difficulties operating below 0°”? However, this is somewhat contradicting the statement at line 293 (negative temperatures registered in June). So, what was the problem then? The operation for extensive time below freezing temperature, or just some component at surface had this issue with temperature (maybe the logger and not the sensor?)?

Line 299: I appreciate the clarity of the figure, though extending the upper limit to 18° or or 20°, to avoid clipping the curves, could be useful (if the peak is not too high), also in view of the white space in page below the figure. Also, check the possible typo in the caption (SFIM is 1450 m or 2450 m?). Also, what does “close to 0°” refer to? It is unclear whether it means that the temp. variations are almost null, or if less variations in temperature are registered when the value is close to 0°. Also, whenever the legend refers to specific point, I suggest referencing to the figure where the elements appear (in general, figure 1c and 1d), to make it easier for the reader to go back and check the location.

Line 304: I assume in b “Moy temp” is the average (“Moyenne”?). If so, please consider changing to “Avg temp”. It can be inferred, but it is better to avoid doubts.

Line 311: “main discontinuity sets” or “main joint sets”.

Line 321: the tern “Main fracture” does not appear in figure 2c. Should this be “Lineament”?

Lines 323-327: In this small paragraph there is the core of the kinematic interpretation of the instability, but I believe this should be expanded (either here or in the discussion, before presenting the scenarios). How should this be expanded? For instance, including a kinematic analysis that will immediately, based on slope orientation and average joint set orientations, what are the feasible failure mechanisms and what are the controlling sets. Such an analysis would make the scenario definition more robust, and would also highlight the role of the various sets in controlling not only the wedge instability, but also the block toppling at the crest. The tab c in figure 14 (particularly the wedge failure sketch) is a nice representation, but also very general, as it does not use the real data, and cannot really be an adequate substitute for a kinematic analysis.

Conversely, the sketch depicting the roto-translational failure is a bit less clear, because the critical factor is not necessarily the scattered orientation of the joints (or sets), but rather, the overall quality of the rock mass (hence, the spacing and persistence of the sets).

Line 344: “Consider” rather than “imagine”

Line 394: Consider changing to “Temperature analysis results”

Line 409: Consider changing to “Rainfall data analysis results” (for consistency)

Line 411: this might be nit picky, but the reference to the periods in figure 12b looks like actual reference to literature. Consider changing [1] and [3] to “1” and “3”. Same goes for other instances.

Line 413: Consider “Conversely” instead of “Contrariwise”

Line 419: Consider “Limitations” instead of “Limits”

Line 427: “Extent” instead of “extend”

Line 426: it is not clear whether this section analyzes the likelihood of the proposed scenario, or a qualitative classification of the impacts generated by the scenario.

Line 430: a very brief description of the chart could be useful for the reader. E.g., what information are used as input (also in terms of type of instability), particularly because the resource might be not readily accessible (it is not at my institution). For instance, my understanding is that the chart provides a qualitative assessment of the likelihood of the investigated scenarios – but I am not sure this is correct.

Line 457: the column “deformation state” should reference to [54]?

Line 458: Based on this chart, I do not understand why the scenarios SS1-SS2 are assigned a deformation state 2-2 and not, for instance, 1-1, where it looks like the description provided in lines 433-434 (displacement at the toe similar to displacement at the top) could also fit. Again, since the resource is not always readily accessible (for me, and I suppose for other scholars as well), it would be worth it to add some details on “how to use” the chart. Also, is there a reason for which the scenario S5 is not (or cannot) be evaluated in this section?

Comments on the Quality of English Language

There are few typos, some of which highlighted in the review

Author Response

Thank you for your enthusiastic feedback and the thorough examination of my article. Your insightful observations and constructive comments were duly taken into account, and we have made every effort to address them comprehensively. Enclosed, you will find the annotated comments and our responses to your review, as well as those from the other reviewers.

Reviewer 3 Report

Comments and Suggestions for Authors

The abstract does not mention any of the data that you will use. It is too generic and needs to be more specific to your case study.

The illustrations are very well done and they significantly enhance the paper.

If Gamma processed the SAR data, why are they not an author on the paper?

I think it is quite original to have the number of different types of data that you have used and for that reason, as well as they eye-catching illustrations, I would publish this paper.

Comments on the Quality of English Language

The English is quite good, but has periodic errors of idiom and grammar that would need to be addressed before publication.  For example, you often say "extend" when you mean "extent", and there are grammatical and / or usage errors in most paragraphs.

Author Response

(The authors gave the same response as above.)

Reviewer 4 Report

Comments and Suggestions for Authors

This article is interesting and provides valuable insights. It focuses on using multiple remote sensing methods that work together to study rock slope instability in the Swiss Alps. With a few minor improvements, it could be suitable for publication. Here are some points to consider:

1. The article's abstract lacks specific detail. It should provide more information, including the methods used and the specific results obtained. If readers only read the abstract, they only found the location of the study case, and not much other useful information.

2. The introduction part is missing some references related to deep-seated rock slope instability.

3. In the introduction part, consider providing a brief overview of remote sensing techniques commonly used in landslides or rock instability studies before diving into the techniques used in your paper.

4. In the Materials and Methods section, specify equipment parameters and cite the software used, such as CloudCompare and Coltop. Additionally, explain how different scenarios were defined, as readers are likely interested in this aspect if they want to use the SLBL in their work.

5. Clarify the term "relative distance" in line 225.

6. Improve the clarity of the description in lines 307-308.

7. Clarify what is meant by "two plans" in line 359. It should be two planes?

8. In Figure 15, ensure that S5 is clearly visible.

9. Section 5.3 of the Discussion seems more like a result than a discussion. An in-depth discussion or interpretation of the relationship between the deformation (rate) and the meteorological factors would be good.

10. The order of sections in the Discussion could be reorganized. Consider moving 5.3 in front of 5.1 and placing 5.4 just after 5.1. There is some uncertainty regarding the necessity of 5.3 in the discussion based on the content presented.

Comments on the Quality of English Language

English writing is OK, but some minor revisions should be made. For instance, "It is possible to define possible scenarios..." in Line 16 in Abstract, seem repetitive. Please check the overall manuscript.

Author Response

(The authors gave the same response as above.)

Reviewer 5 Report

Comments and Suggestions for Authors

It is an interesting contribution regarding various approaches of studying a complex landslide and it concludes on scenarios. So, by definition is interesting. However, in some cases, see the attached annotated version, the presentation is rather vague and the description unclear. So my judgment is major revision.  

Comments on the Quality of English Language

The written English is at a rather good quality, but it will valuable a careful editing of the text.

Author Response

(The authors gave the same response as above.)

Round 2

Reviewer 2 Report

Comments and Suggestions for Authors

Dear authors,

thank you for the effort of addressing the comments that were made in the first round of review, and for explaining the challenges that made it impossible to address the one related to the change detection analysis.

I believe the manuscript is now ready for publication - the only minor thing that I recommend changing is related to the Markland's test. Specifically, authors used two values for friction angle, 0° and 30°. I would really recommend removing the analysis for the 0° friction angle: on the one hand it addresses the kinematic feasibility of the investigated mechanism. On the other hand, a 0° friction angle is not realistic even for wet or filled fractures, as it would mean that sliding would occur for planar discontinuities. I suggest either leaving the only 30° analysis or, even better, to use a more realistic value, say 15°, to represent the less favorable conditions mentioned in the manuscript - water pressure and clay infill in fractures.

Also, making the results of the Markland's test less "transparent" would improve the readability of the figure.

All the best.

Author Response

Dear Editor and reviewers,

Thank you for the fast second revision of the manuscript. The last suggested improvements mentioned by reviewer 2 was taken into account:

  • The text in the methodology section was changed to: A low friction angle of 15° was used to account for potential pore pressure and the filling of the fractures with fine materials. (line 226),
  • Figure 15 was modified with the results for a friction angle of 15°. The area susceptible to instabilities are less transparent, as suggested.
  • The additional figure S1 is also modified to account for those modifications.

We are grateful for the insightful comments that have undoubtedly enhanced the quality of our manuscript. We look forward to the publication of the article.

Sincerely, and on behalf of all co-authors,

Charlotte Wolff

Reviewer 5 Report

Comments and Suggestions for Authors

No more comments

Author Response

(The authors gave the same response as above.)
